# A Global Sharing Mechanism of Resources: Modeling a Crucial Step in the Fight against Pandemics

**DOI:** 10.3390/ijerph19105930

**Published:** 2022-05-13

**Authors:** Katinka den Nijs, Jose Edivaldo, Bas D. L. Châtel, Jeroen F. Uleman, Marcel Olde Rikkert, Heiman Wertheim, Rick Quax

**Affiliations:** 1Computational Science Lab, Faculty of Science, University of Amsterdam, 1098 XH Amsterdam, The Netherlands; jose.edivaldo.fisica@gmail.com (J.E.); r.quax@uva.nl (R.Q.); 2Radboud University Medical Center, Department of Geriatric Medicine, 6500 GL Nijmegen, The Netherlands; bas.chatel@radboudumc.nl (B.D.L.C.); jeroen.uleman@radboudumc.nl (J.F.U.); marcel.olderikkert@radboudumc.nl (M.O.R.); 3Institute for Advanced Study, 1012 GC Amsterdam, The Netherlands; 4Radboud Center for Infectious Diseases, Department of Medical Microbiology, Radboud University Medical Center, 6525 GA Nijmegen, The Netherlands; heiman.wertheim@radboudumc.nl

**Keywords:** COVID-19, resilience, supply chain, collaboration, personal protection equipment

## Abstract

To face crises like the COVID-19 pandemic, resources such as personal protection equipment (PPE) are needed to reduce the infection rate and protect those in close contact with patients. The increasing demand for those products can, together with pandemic-related disruptions in the global supply chain, induce major local resource scarcities. During the first phase of the COVID-19 pandemic, we witnessed a reflex of ‘our people first’ in many regions. In this paper, however, we show that a cooperative sharing mechanism can substantially improve the ability to face epidemics. We present a stylized model in which communities share their resources such that each can receive them whenever a local epidemic flares up. Our main finding is that cooperative sharing can prevent local resource exhaustion and reduce the total number of infected cases. Crucially, beneficial effects of sharing are found for a large range of possible community sizes and cooperation combinations, not only for small communities being helped by large communities. Furthermore, we show that the success of sharing resources heavily depends on having a sufficiently long delay between the onsets of epidemics in different communities. These results thus urge for the pairing of a global sharing mechanism with measures to slow down the spread of infections from one community to the other. Our work uses a stylized model to convey an important and clear message to a broad public, advocating that cooperative sharing strategies in international resource crises are the most beneficial strategy for all. It stresses essential underlying principles of and contributes to designing a resilient global supply chain mechanism able to deal with future pandemics by design, rather than being subjected to the coincidental and unequal distribution of opportunities per community that we see at present.

## 1. Introduction

To keep a pandemic like the one caused by COVID-19 [1] under control, resources such as personal protection equipment (PPE; i.e., face masks, gloves, and disinfectants) are critical both for the population and for health care personnel [2]. Unfortunately, the demand for PPE increases exponentially with the spread of a new epidemic. As the available resource stock is tailored to the population’s standard use, this poses serious challenges to the associated resource supply chains. Consequently, several countries with high infection rates reached a state of low to no resource availability during the various waves of the COVID-19 pandemic, irrespective of their gross domestic products (GDP) [3,4].

Due to an exponential increase in demand during epidemic waves, local production increases are rarely fast enough to cover the high need [5]. Moreover, local production is not the primary provider in many countries that obtain PPE from abroad. In some places, preventive methods were proposed to mitigate the upcoming shortages [6], such as setting guidelines for rational and safe prolonged PPE use or reuse [7] and alternative solutions for protection [8]. Such mitigation strategies may provide temporary relief but do not address the bigger scalability problem. Moreover, some of these strategies lead to healthcare workers feeling inadequately protected and being fearful of getting infected [9].

The primary production of medical products comes from global supply chains (GSC). GSCs have been optimized for cost-efficiency and may therefore lack resilience, such as redundant supply lines or large buffers along the chains. For this reason, a major topic of debate is if and how these GSCs should be restructured or replaced [7,10,11]. An example of GSC fragility is the disruption of several technological- and health-product factories in Wuhan during the COVID-19 pandemic and the closing of shipping ports, impacting the international market and availability of these products on a very short time scale [12,13].

Consequently, due to the resource scarcity threat in the first wave of the COVID-19 pandemic, multiple countries started to stock, confiscate, and prohibit the exportation of PPE [14,15] and diagnostic testing materials. This affected the global market as a whole since no country is fully self-sufficient in all the different types of PPE. Furthermore, it greatly impacted low-income countries that had to deal with increased prices and a scarcity of goods [14,15,16,17]. Although a plausible solution would be to redesign the GSCs for increased systemic resilience against rare incidents, such as pandemics, the restructuring of infrastructure may very well pose a prohibitive cost. Additionally, it would require a globally concerted effort at an unprecedented scale.

Instead, this paper investigates the question of whether implementing a resource-sharing policy among different regions or countries can enhance essential-resource stock resilience. A PPE-sharing policy differs from increasing the overall resilience of GSCs: instead of ensuring the transport lines from producers to each individual consumer region, a sharing policy entails creating flexible transport lines between the consumer regions. For this reason, it may provide a more feasible approach that is effective at suppressing future pandemics. The current research field into epidemic spreading and COVID-19 lacks studies on sharing strategies for sharing this type of resource. However, the effects of vaccine sharing between two populations have already been studied using a classical multi-compartmental susceptible–infected–recovered (SIR)-based model [18], concluding that overall health outcomes can greatly benefit from cooperative sharing [19]. Although both vaccine and PPE availability are important factors in successfully mitigating a pandemic, they are, respectively, aimed at pre-epidemic immunization increase and within-epidemic disease containment. It is therefore important to study the effects of PPE sharing separately. A convincing point to share resources used during epidemics was made using a stochastic model considering ventilator allocation [20]. However, the explicit disease transmission dynamics within populations could not be taken into account by this approach. Because so far, concrete studies of PPE-resource-sharing effects on these transmission dynamics are thus absent, we want to create an SIR-inspired multi-compartmental model to study this specifically.

In this article we will first present the stylized model we created to investigate cooperative resource sharing, followed by a simulation of sharing scenarios between different GDP-sized community combinations. In brief, the model concerns two communities that can be interpreted as countries, regions, or even cities. We modeled their local epidemics and the spread of infections across the communities, in combination with their (unspecified) PPE resources. PPE availability lowers the infection rate by offering protection and therefore enables suppressing the infection spread to some degree. Of interest are the effects of simulating changes in resource-sharing behavior on the capacity of going through the entire epidemic without fully depleting the PPE stock.

The specific contribution of this study is thus not the novelty of our modeling techniques, but presenting an important message to a broad public by using a stylized model. Motivated by the non-cooperative strategies observed at present, it has the purpose to understand at a high level the impact of sharing essential resources and which factors should be taken into account to reach a better local situation for all communities involved.

## 2. Methods

Here follows a functional description of the model needed to understand the results. Readers interested in more technical details are referred to the Extended Methods in Appendix A. Here, Table A1 can also be found with an overview and justification of the assumed model parameters.

### 2.1. Coupled SIR and DSP Model

We developed a simplified demand–stock–production (DSP) model to emulate the usage and production of PPE. PPE can be interpreted here as any physical equipment that may help limit disease transmission. To cover the disease dynamics, including the transmission-mitigating effects of PPE, the DSP model is coupled to a classical susceptible–infected–recovered (SIR) model [21,22] without births and deaths. As we model the occurrence of one epidemic wave per community at most, we assume a total simulation duration short enough to justify this simplification. N.B.: throughout this paper the letter *S* will be used to refer to the PPE ‘stock’ from the DSP model, as susceptibles (from the SIR model) are never referred to outside of the appendix.

For the inter-model coupling, the following two assumptions are used: (1) more protective equipment is needed when the amount of infected increases [3,9]; and (2) the unavailability of PPE increases the infection rate of the disease [4,5]. This leads to the following two connecting factors: (1) a community’s current PPE demand in the DSP chain depends directly on their number of infected and an assumed constant of necessary PPE per infected case; and (2) the effective reproduction number $R_{e}$ of the SIR model depends on the available PPE stock. Concretely, this means that in absence of PPE, $R_{e}$ is set to the standard reproduction number $R_{0}$ of the virus. In presence of PPE, the $R_{e}$ is reduced to a lower value with $R_{e}<1$. In the appendix, explicit calculations and justifications of the assumed constant values are found. An important assumption made in this model implementation is thus that the $R_{e}$ only reduces to $R_{0}$ upon full PPE stock depletion, a point we consider further in the discussion.

In all experiments, two communities *A* and *B* of *n* individuals each are simulated. They differ only in maximum stock ($S^{\max}$) and epidemic onset time. The results are described from the point of view of community *A*. Therefore, we assume epidemic onset in *A* to always be at t=0, with the introduction of one infected individual into this community. The epidemic onset in *B* is at a certain time difference Δt from t=0, thus at t=Δt. This time difference models the time it would take for the epidemic to spread between the two communities directly, or via another (‘confounding’) community. The difference can be Δt=0 days for simultaneous onset and take on both positive and negative values. In this way, both an epidemic starting in *A* followed by a spread to *B* and vice versa can be investigated. This enables us to study the effects of changes in the time difference of the epidemic onset between *A* and *B* and in their relative order. The differences in maximum stock reflect the different capacities to directly access the necessary PPE, due to, e.g., differences in GDP and storage capacity. By using different maximum stock levels we can investigate how possible asymmetries between communities might impact sharing behavior outcomes.

### 2.2. Sharing System

We further implemented a PPE-sharing system between the two communities to simulate sharing behavior. Sharing is managed via a switch mechanism that depends on the ratio between the current and maximum stock of each community $S/S^{\max}$ and a sharing threshold θ. The sharing threshold is a critical stock ratio agreed upon beforehand by the communities. We assume the communities want to ensure always having at least this proportion of stock available to themselves, to increase their chances of weathering a local epidemic without reaching stock depletion (S=0). If during an epidemic the stock of a community falls below the sharing threshold θ, they request support from the other community. The switch mechanism activates the sharing system if the other community has the possibility to share, i.e., if their own stock ratio is above the sharing threshold. If the second community starts sharing, they are assumed to continue until reaching the sharing threshold θ themselves or until the other community’s stock is restored to at least θ again. In summary, the switch mechanism leads to active PPE sharing if and only if exactly one of the communities’ stock levels is below the sharing threshold and the other community has a stock level above the threshold. A low sharing threshold θ corresponds to communities being willing to share a large fraction of their stock with one another, while a high θ indicates that they intend to keep a large fraction of stock for themselves.

## 3. Results

The effects of differences in the relative time difference between communities’ epidemic onset, maximum stock height, and sharing threshold were assessed in different scenarios that are outlined below. We will first discuss the case where the collaborating communities have equal maximum stocks $S^{\max}$ and then scenarios where these are unequal.

Our main results are shown in the form of phase plots with a distinctive color scheme. We discriminate between three qualitative outcomes in the phase plots, shown in red, white, and blue, in decreasing order of severity. In all phase plots, the total infection incidence is given as the ratio of individuals within the populations that has been infected during the disease spread: the infected ratio. If the stocks of both communities would deplete fully at any point during the epidemic, then their effective infection reproduction numbers would increase enormously. This would lead to the highest possible total infected ratio, corresponding to the red state. If only one community would reach zero stock then this would lead to an intermediate total infected ratio, corresponding to the white intermediate state. Finally, if both communities could weather the epidemic without reaching zero stock, the blue state would be reached with the lowest possible infected ratio. Intermediate transition phases, i.e., those visible in orange, could appear due to small differences in the total infected ratio, for example due to slight differences in the zero-stock-period duration.

### 3.1. Equal Maximum Stocks

We can observe that sharing between certain equal maximum stock communities can prevent stock depletion and the associated high infected ratio. Figure 1 shows typical epidemic time evolutions for two equal maximum stock communities and the difference between non-sharing and sharing. This clearly depicts how non-sharing (left column) for these communities leads to stock depletion. During a stock depletion period (indicated by the red shading) $R_{e}$ becomes larger and therefore leads to stronger exponential growth and a high peak of infections. In the sharing condition (right column) we can see that stock sharing with the community in need can prevent full stock depletion and thus prevent the $R_{e}$ increase, keeping the associated incidence peak to a manageable level. This ultimately leads to a significantly lower total of infected cases in both communities (area under the infected curves).

The size of the time difference (Δt) between epidemic onsets heavily influences if communities as the ones shown here can reach low infected ratios for one or even both communities by sharing their stock. This is illustrated by the middle pane of Figure 2 for medium-stock communities (defined to have a maximum stock of $S^{\max}=3\times 10^{7}$). When not sharing (θ=1.0) these communities end up in the red state, both having the highest infected ratio. However, if the sharing threshold is low enough and there is some time difference between the epidemic onsets, the communities can reach the white state in which only one of the two communities (*B* in this case) reaches stock depletion. Although this is already a successful sharing example for *A*, the time difference is still too short for the combined stocks to be restored sufficiently for the second epidemic in *B*. That being said, we can see that when the time difference between *A* and *B* epidemic onsets is increased further, both communities can achieve a low infected ratio (blue state). In the latter case, there was enough time to replenish the stock levels between the consecutive epidemics to save both communities.

We defined the medium stock category with this maximum stock of $S^{\max}=3\times 10^{7}$ to explicitly show the above described behavior when cooperating with a community that is equal in maximum stock. The other distinctive categories we use throughout this paper are low-stock ($S^{\max}=1\times 10^{7}$) and high-stock ($S^{\max}=7\times 10^{7}$) communities. Low stock is set at this specific level as these communities show high infected ratios (red state) individually and for all possible conditions when sharing with another low-stock group, as can be seen in the left pane of Figure 2. Conversely, high-stock communities always reach low infected ratios (blue state in the right pane) irrespective of their sharing behavior with an equal-stock community. As mentioned before, virtually no actual countries exist where (all possible) PPE stock levels can be high enough to endure a whole epidemic of this sort. The high stock category is therefore included merely for illustrational purposes.

The observation that sufficient time difference increases the chance of having one or both sharing communities reach a low infected ratio also holds for more intermediate maximum stock communities. This becomes apparent from Figure 3, which shows sharing between more intermediate maximum stock sizes for three time-difference categories that can be seen as cross-sections (at Δt=0, Δt=60 and Δt=180) of the phase plots in Figure 2 and vice versa (at the maximum stock categories, see green dashed lines). We can observe that equal-stock communities can already benefit from sharing from a maximum stock as low as $S^{\max}\approx 2\times 10^{7}$ for big enough time differences. We also see that the lower the maximum stock, the more important implementing a low sharing threshold is for sharing success. The white area in which community *B* (the community that gets infected secondly) still reaches high infected numbers even diminishes for the lower maximum stock communities when Δt=180.

In summary, the most important result for equal maximum stock community sharing is that communities with insufficient stock to weather the epidemic by themselves can, when agreeing on sufficiently low sharing thresholds, pass the epidemic without reaching the maximal infected ratio. Crucially, for these communities, a large-scale epidemic can be averted only for either or both of them when the time difference between epidemic onsets is sufficiently long. This requires a tailored strategy depending on the dynamics of the disease at hand, but always with a strong focus on mitigating disease spread between different regions by, for example, imposing travel restrictions early in epidemic outbreaks.

### 3.2. Unequal Maximum Stocks

Upon reading the equal maximum stock results, one could start fearing that sharing poses a risk to the second community. However, as we will further show by presenting the unequal maximum stock sharing results, this fear is unjustified for two reasons: (1) for a low- or medium-stock community, sharing never leads to worse results than non-sharing for both equal and unequal maximum stock combinations; and (2) for higher-stock communities there is hardly any risk in unequal sharing-even when the total shared stock turns out to be insufficiently high for both communities, a higher-stock second community can still weather the epidemic with a low infected ratio (Figure 4).

Only for very specific low time differences between epidemic onsets, a secondly affected higher-stock community could possibly be worse off, i.e., experience so-called ‘second community disadvantage’ (in the white areas indicated with *B* in the figure). This should again stress the importance of implementing measures to sufficiently slow down infection spread between communities, rather than discouraging higher-stock communities from sharing. Furthermore, in Figure 4c, we can even observe that a low enough sharing threshold fully abolished the chance for high-stock communities of having the second community disadvantage. However, as this was only the case for these very specific disease dynamics and stock levels, communities should not solely rely on a low sharing threshold and still see increasing the onset time difference as a top priority. The substantial reduction of the second community disadvantage for higher-stock communities is a major difference from what we have seen for equal community sharing, in which the white state always meant full depletion for the community affected secondly by the epidemic.

An additional interesting observation is that sharing between unequal communities could help both communities weather the epidemic for all maximum stock height combinations. Even the low- and medium-stock communities can together end up in blue states by sharing! To reach blue states, the ordering of epidemic onsets is not of major importance, as we can see from the approximate similar sizes of the blue states for both positive and negative time differences. What is of major importance to obtain from these results, is (again) the time differences between these onsets and agreeing on a low sharing threshold.

In other aspects however, the epidemic onset ordering does impact sharing outcomes, which is indicated by some asymmetries that can be observed in the phase plots around Δt=0: an epidemic that starts in the highest-stock community (negative Δt’s) will in general less often lead to the highest total infected ratios (red and orange states) than vice versa. It should be noted, however, that the lowest infected ratio regions (blue states) are often bigger when the lower-stock community is infected first (positive Δt’s). If starting in the lower-stock community, we see that increasing the time difference between onsets is of utmost importance to avoid high total infected ratios. The longer this difference between the epidemics, the more time for stock regeneration.

An interesting illustration of the effects of increasing the time difference between onset in a lower-stock and higher-stock community is shown in Figure 5, for which the relevant sharing threshold and time-difference settings were indicated in Figure 4 with the pink dashed line.

By increasing the threshold, the system passes from a state of depleted stock for the first community, via the maximal total infected ratio and a state of second community disadvantage, to both communities reaching low infected ratios. This effect is indeed asymmetric around Δt=0 if time evolutions would be taken for the same threshold, but with negative Δt’s (the mirror-image of the pink dashed line), we would only observe the state in which the lower-stock community depletes fully. We can further observe that the asymmetry in the phase plots vanishes in the larger time-difference regions. This is because, after sufficient time, the first epidemic dynamics die out and the stocks replenish enough, resulting in virtually no interaction effects between the consecutive epidemics.

The results discussed here solely focus on the three stock categories, as we defined before, to ease result interpretation. For more intermediate stock values, we refer to Figure A1 in Appendix B, where the effect of an increasing $S_{B}^{\max}$ on infected ratio is shown for different fixed $S_{A}^{\max}$ and different onset-time-difference categories.

Summarizing, the unequal maximum stock sharing results presented here again stress the positive combined effect of a low sharing threshold and a sufficient time difference between epidemic onsets. Additionally they show (1) that there is no downside from sharing for lower-stock communities and (2) how high-stock communities may substantially improve the overall epidemic resilience for both communities combined without much increased risk for their own population, as long as sufficient measures are implemented to reduce epidemic spread between the communities.

## 4. Discussion

We have introduced a stylized PPE production model with a sharing mechanism for communities facing an epidemic. The model convincingly shows that involved PPE stock sharing can greatly improve total and local epidemic resilience for all combinations of maximum stock communities. Crucially, these sharing strategies should be accompanied by disease spread mitigation measures to increase the time between epidemic peaks. This ensures that both communities have access to large quantities of stock when they need it most. First, we saw that completely blocking the circulation of essential supplies can lead to local stock shortages and subsequently epidemic proliferation. Then, we showed that for all realistically possible scenarios (where communities never have enough stock to fully endure exceptional situations, such as the COVID-19 pandemic, by themselves), sharing leads to better overall and local epidemic outcomes than non-sharing. Even if a high-stock community would exist, sharing can never lead to a more negative outcome for any cooperating community as long as the epidemic onset time differences are long enough. It is good to additionally realize that investing in the global reduction of infection rates helps to restrict the total viral circulation and the chance of reinfection in each region. Therefore, these investments are beneficial to all and crucial for final pandemic mitigation. The results thus strongly advocate for a willingness to share big stock proportions, with a special focus on accompanying measures to slow down epidemic spread between communities. In this way, pandemics can be fought way more efficiently and successfully for all parties involved with their respective PPE production limitations.

We emphasize that the numbers used in the examples are not intended to be realistic and are merely for illustrational purposes. In particular, what time difference would be ‘sufficient’ will depend on the properties of the virus, the modes in which people have contacts with other people, the infrastructure and production capabilities of the PPE, the preventive measures in place, and so on.

As mentioned before: increasing the delay between epidemic onset should evidently be a major focus when implementing this sharing mechanism. To achieve this, participating countries must directly focus on, e.g., limiting international travel when a community gets infected. Interestingly, the process of resource sharing itself could then start posing a threat, by increasing the chance of importing infection cases [23]. This could limit the beneficial effects of implementing the proposed sharing method if not attended to well enough. We believe, however, that setting clear safety guidelines for this process can lower the risk sufficiently to still make sharing the most viable option. For interested policymakers, the model could be extended with the possibility to import infection cases during sharing. In this way, this possible risk and the necessary precautions could be investigated in more detail.

There are some simplifications in our model, which could possibly be accounted for in further research. For example, the effective reproduction number $R_{e}$ only changes when the community’s stock fully depletes (S=0). In a more realistic scenario, there would be a gradual increase in $R_{e}$ as the stock becomes more scarce and less people can be protected. This would result in smoother phase plots, abolishing the three clear outcome states observed in this paper. Nevertheless, the general result that sharing reduces the infected ratio will then still hold. This more realistic approach would therefore make the model interpretations more complex without adding new insights. Further, parameters such as the basic reproduction number $R_{0}$, the population size *n*, and PPE production rate *P* could differ substantially between communities. At the moment, the only characteristic we varied between communities was their maximum stock $S^{\max}$, but it could be interesting to study the effects of greater heterogeneity between communities on the results. One last simplification worth mentioning is the instantaneous transfer of huge PPE stock quantities during sharing. In real life, infrastructure limitations could form a barrier to implement the proposed sharing strategies by slowing down the redistribution process and limiting sharing effectiveness. Although it was beyond the scope of this paper, it could thus be interesting to extend the model with some geographical transportation limits and delays for different distances to ascertain the boundary conditions for sharing to be effective.

Further interesting extensions could be to include more communities and to tailor the DSP model to specific and possibly multiple resource types. When multiple communities are involved, the necessary individual sharing amount to sufficiently help a community in need reduces as the load can be divided over contributing groups. In our model we have seen that two low-stock communities had too little combined stock to save either of them, even when sharing major proportions of stock. But a bigger group of solely low-stock communities could potentially have enough combined stock to successfully avoid high infected ratios. However, one should note that in this more complex scenario, a more complex sharing strategy decision structure also arises. To model the politics of this, game-theoretic approaches could be used [24,25]. For the extensions suggested here, the modeler should further be aware of the increasing computational costs of solving 3(n+r+1) ODEs for *n* communities and *r* resource types. Lastly, in all its simplicity, our model could serve as a basis for models of higher complexity that more closely resemble real-world scenarios. After fitting the data of manufacturers, resource usage, and epidemic dynamics, optimal sharing strategies could potentially be estimated with some precision for combinations of collaborating communities, depending on the uncertainties involved. A similar work of fitting is done to predict SARS-CoV-2 evolution [26].

Overall, we can see that increasing global resilience to future pandemics is a complex problem that requires an interdisciplinary effort. Although myriad extensions could be made to this model, these additions would have unnecessarily distracted the reader from our main aim of communicating to policymakers how cooperation can improve epidemic resilience. The strengths of this model are its simplicity and comprehensibility, conveying a clear message and recommendations.

## 5. Conclusions

In summary, this work demonstrates how trade barriers for essential healthcare resources can worsen pandemic outcomes, while cooperation between different communities can successfully avoid resource scarcity and prevent high-infected incidences for all parties involved. We used a stylized model encompassing disease spread and a resource supply chain in two communities. If enough measures were taken to slow down infection spread between them, both communities were able to avoid resource exhaustion at any point during their epidemics by cooperative sharing. This shows that collaboration is a viable solution to avoid resource scarcity. This message is similarly true for vaccines, as we have learned during the current COVID-19 pandemic [19].

The model serves as a proof-of-concept that illustrates how sharing protective equipment resources between communities is a crucial step in epidemic mitigation. Although the model is stylized, we believe that its general insights are easy to understand and essential to disseminate to all stakeholders involved. It advocates for involved sharing strategies between all types of communities, showing that with a high willingness to share and a strong focus on slowing international infection spread, total epidemic mitigation outcomes can be improved greatly. It also successfully demonstrates that high GDP countries should not be afraid to offer help to countries that need it, as potential negative effects for their own population can be abolished if the right disease spread mitigation measures are in place. Furthermore, reducing viral circulation globally reduces each region’s chance of reinfection and is thus in the interest of all. By taking these insights into account, we believe that global initiatives can help prevent essential-resource exhaustion during pandemics. Our study thus shows that mutual resource sharing can potentially support all participating communities during their local epidemics and ultimately lead to the collaborative mitigation of a pandemic.

## Figures and Tables

**Figure 1 ijerph-19-05930-f001:**
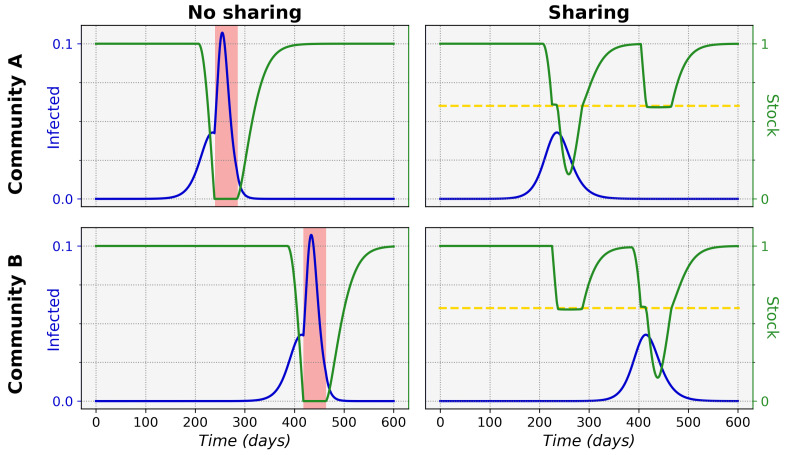
**Time evolution of epidemic development in two communities without and with sharing mechanism.** Shown are the proportions of infected (blue) relative to the total community population and the proportions of stock (green) relative to the maximum stock over time. The communities had an equal maximum stock of $S^{\max}=3\times 10^{7}$. The only difference between the communities was epidemic onset, with epidemic introduction in *A* at t=0 and in *B* at t=Δt=180 days. Critical zero-stock periods are shaded in red in the no-sharing condition (left column). The lack of stock leads to an increase in epidemic spread and a high infected peak. The sharing threshold θ=0.6 (yellow dotted line) is indicated in the sharing condition (right column). Here, sharing prevents the total depletion of stock and the associated high infected peak: both the local and total infected ratios are low. $n=1.7\times 10^{7}$, $R_{0}=2.3$, γ=1/6, r=0.4, w=4, $P\left(0\right)=5\times 10^{7}$, $P^{\max}=4\times P\left(0\right)$, D(0)=P(0).

**Figure 2 ijerph-19-05930-f002:**
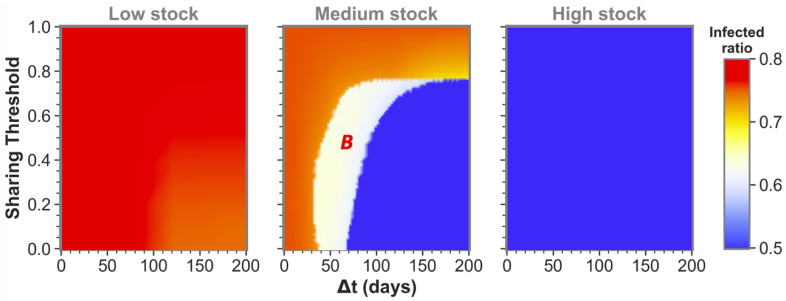
**Influence of epidemic onset time difference and sharing threshold on total infected ratio for equal maximum stock communities.** The epidemic is introduced in *A* at t=0, and in *B* at t=Δt. The indicated infected ratio is the average of the two communities’ infected ratios. The red state indicates both communities reaching a high infected ratio and white means that one community (labeled with the red letter, here *B*) had a high infected ratio while the other remained low. In the blue state both communities had a relatively low amount of infected. For each phase plot, the two communities had the same maximum stock, being either $S^{\max}=1\times 10^{7}$ (Low stock), $S^{\max}=3\times 10^{7}$ (Medium stock) or $S^{\max}=7\times 10^{7}$ (High stock). $n=1.7\times 10^{7}$, $R_{0}=2.3$, γ=1/6, r=0.4, w=4, $P\left(0\right)=5\times 10^{7}$, $P^{\max}=4\times P\left(0\right)$, D(0)=P(0).

**Figure 3 ijerph-19-05930-f003:**
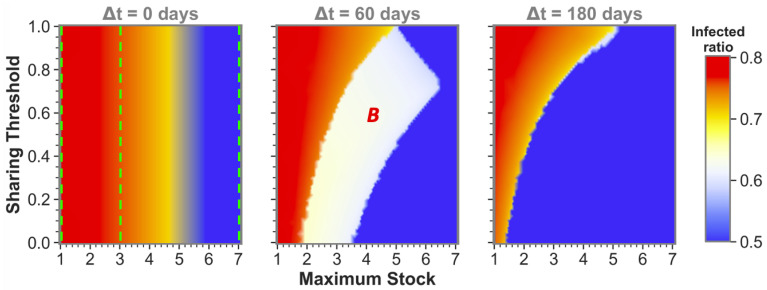
**Influence of maximum stock size and sharing threshold on the total infected ratio for three distinct epidemic onset time differences between equal maximum stock communities.** In each simulation, the two communities had equal maximum stock sizes. The white state is labeled with the community that reached zero stock and a high infected ratio (*B*). The green dashed lines in the left panel indicate where cross-sections are taken for the three maximum stock categories used throughout this paper: Low stock with $S^{\max}=1\times 10^{7}$, Medium stock with $S^{\max}=3\times 10^{7}$ and High stock with $S^{\max}=7\times 10^{7}$. The values at these lines correspond to those at the different maximum stock phase plots in Figure 2 for Δt=0. Within the Δt=60 and Δt=180 phase plots, cross-sections at the same stock values can be imagined for likewise comparison with the previous figure. $n=1.7\times 10^{7}$, $R_{0}=2.3$, γ=1/6, r=0.4, w=4, $P\left(0\right)=5\times 10^{7}$, $P^{\max}=4\times P\left(0\right)$, D(0)=P(0).

**Figure 4 ijerph-19-05930-f004:**
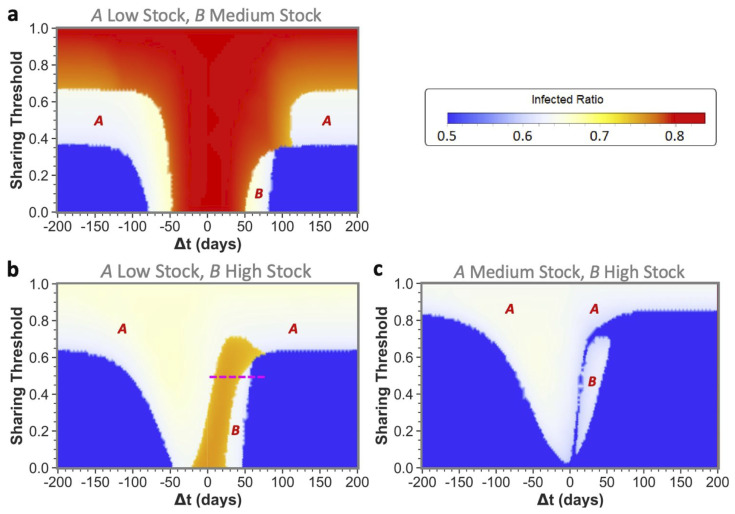
**Influence of time difference between epidemic onset and sharing threshold on total infected ratio for unequal maximum stock communities.** Respectively showing combinations of (**a**) low- and medium-stock (**b**) low- and high-stock and (**c**) medium- and high-stock communities. The epidemic is introduced in *A* at t=0, and in *B* at t=Δt. In the white areas, a red letter indicates which of the two communities reached zero stock. The pink dashed line in (**b**) indicates the settings taken for the time evolutions shown in Figure 5. $n=1.7\times 10^{7}$, $R_{0}=2.3$, γ=1/6, r=0.4, w=4, $P\left(0\right)=5\times 10^{7}$, $P^{\max}=4\times P_{i}\left(0\right)$, D(0)=P(0).

**Figure 5 ijerph-19-05930-f005:**
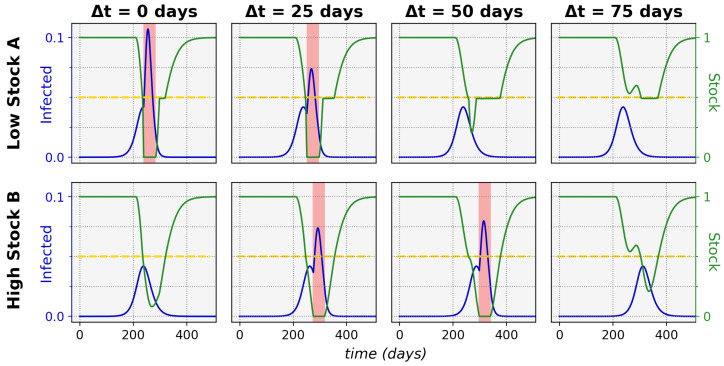
**Effect of increasing epidemic onset time difference on sharing success of unequal communities.** Indicated are the proportions of infected (blue) relative to the total community population and the proportions of stock (green) relative to the maximum stock over time. Critical zero-stock periods are shaded in red and the sharing threshold (θ=0.5) is indicated by the yellow dotted line. The communities had unequal maximum stocks, with $S_{A}^{\max}=1\times 10^{7}$ and $S_{B}^{\max}=7\times 10^{7}$. By increasing the epidemic onset time difference, the system transitions from only high-stock community *B* weathering the epidemic with low infected ratios (for Δt=0 days), via both communities reaching high infected ratios (for Δt=25 days) and only low-stock community *A* reaching a low infected ratio (for Δt=50 days), to both having low infected ratios (for Δt=75 days). This behavior is observed for the specific settings, as indicated by the pink dashed line in Figure 4, but more generally illustrates the impact of epidemic onset time difference on the possible system outcomes. $n=1.7\times 10^{7}$, $R_{0}=2.3$, γ=1/6, r=0.4, w=4, $P\left(0\right)=5\times 10^{7}$, $P^{\max}=4\times P\left(0\right)$, D(0)=P(0).

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
