# Peer review of "A Global Sharing Mechanism of Resources: Modeling a Crucial Step in the Fight against Pandemics"

_ijerph, 2022, doi:10.3390/ijerph19105930_

Round 1
Reviewer 1 Report
- Studying the resources control during a pandemic in terms of sharing mechanism, is an interesting and worthy research topic. However, the research assumptions and experiments need to be explained and justified in a more rigorous manner.
- There is no discussion grounded to justify the two connecting factors, including current PPE demand in the DSP chain and the effective reproduction number Re of the SIR model, based on the two assumptions (p3).
3. In a similar sense, there is no rationale has been provided as to the two experimental units, isolated communities A and B, independence. (p3).
Reviewer 2 Report
In this paper, besides showing that a cooperative sharing mechanism can substantially improve the ability to face epidemics, the authors present a stylized model in which communities share their resources such that each can receive resources whenever a local epidemic flares up. They also show that the success of sharing resources heavily depends on having a sufficiently long delay between the onset of epidemics in different communities. This means that a global sharing mechanism should be paired with measures to slow down the spread of infections from one community to the other. Their work constitutes a first step towards designing a resilient global supply chain mechanism that can deal with future pandemics by design, rather than being subjected to the coincidental and unequal distribution of opportunities per community at present.
The paper is well written and the results seem correct. Nevertheless, a few points must be addressed to the authors:
1- In the first place, I would encourage the authors to extend the abstract more with the key results. As it is, the abstract is a little thin and does not quite convey the interesting results that follow in the main paper. The ”Abstract” section can be made much more impressive by highlighting your contributions. The contribution of the study should be explained simply and clearly. Also, the abstract must do not contain citations.
2- The Introduction section needs a major revision in terms of providing a more accurate and informative literature review pros and cons of the available approaches and how the proposed method is different comparatively. Also, the motivation and contribution should be stated more clearly.
3- The authors must justify the fact that they use the classical SIR ordinary differential equation formulation without births or deaths. This simplification can be justifiable when the epidemic duration is very short. For example, in the context of the Covid-19 pandemic, this simplification was obvious at the beginning of the pandemic. However, after 3 years, all models of Covid-19 include births or deaths. This is why it is very important to justify the simplicity of the proposed model.
Reviewer 3 Report
Having completed the review of the paper "A global sharing mechanism of resources: modeling a crucial step in the fight against pandemics" I identified an interesting topic and valuable research, however I suggest below some aspects:
Introduction:
This item is necessary to rewrite I consider that it is very succinct, because it is not clear the purpose of the study, as is the current situation of the field of research, I do not identify the hypothesis raised or the question you want to answer. It is important to give the reader a brief context of how the document will play out.
Discussion
I suggest including in the conclusions what are the barriers/limiting factors the authors identified to implement the proposed method?
Conclusions
I suggest summarizing the conclusions, these should be concise and clear.
Reviewer 4 Report
I think this is a good paper, very clear and, methodologically substantial.
in line 71 and 72 Here follows a functional description of the model needed to understand the results. Readers interested in more technical details are referred to the Extended Methods in Appendix A. But I did't find it in your manuscript.
Round 2
Reviewer 2 Report
The manuscript is well improved. I recommend it for publication